# Effects of Stress-Relieving Temperature on Residual Stresses, Microstructure and Mechanical Behaviour of Inconel 625 Processed by PBF-LB/M

**Alessandra Martucci** [1,2,*] **, Giulio Marchese** [1,2,3] **, Emilio Bassini** [1,2,3] **and Mariangela Lombardi** [1,2,3]

1   DISAT–Department of Applied Science and Technology, Politecnico di Torino, Corso Duca degli Abruzzi 24, 10129 Turin, Italy
2   IAM@PoliTo–Integrated Additive Manufacturing, Corso Castelfidardo 51, 10129 Turin, Italy
3   Consorzio Interuniversitario Nazionale per la Scienza e Tecnologia dei Materiali (INSTM), Via G. Giusti 9, 50121 Florence, Italy
*   Correspondence: alessandra.martucci@polito.it

**Abstract:** Inconel 625 (IN625) superalloys can be easily fabricated by the laser-based powder bed fusion (PBF-LB/M) process, allowing the production of components with a high level of design freedom. However, one of the main drawbacks of the PBF-LB/M process is the control over thermally induced stresses and their mitigation. A standard approach to prevent distortion caused by residual stress is performing a stress-relieving (SR) heat treatment before cutting the parts from the building platform. Differently from the cast or wrought alloy, in additively manufactured IN625, the standard SR at 870 °C provokes the early formation of the undesirable δ phase. Therefore, this unsuitable precipitation observed in the PBF-LB/M material drives the attention to develop a tailored SR treatment to minimise the presence of undesirable phases. This work investigates SR at lower temperatures by simultaneously considering their effects on residual stress mitigation, microstructural evolution, and mechanical properties. A multiscale approach with cantilever and X-ray technologies was used to investigate how the residual stress level is affected by SR temperature. Moreover, microstructural analyses and phase identifications were performed by SEM, XRD, EBSD, and DSC analyses. Finally, mechanical investigations through microhardness and tensile tests were performed as well. The results revealed that for the additively manufactured IN625 parts, an alternative SR treatment able to mitigate the residual stresses without a massive formation of δ phase could be performed in a temperature range between 750 and 800 °C.

**Keywords:** Ni-based superalloys; IN625; residual stress; stress-relieving treatment; laser-based powder bed fusion; additive manufacturing





## 1. Introduction

Ni-based superalloys have attracted interest in aerospace and automotive applications thanks to their excellent fatigue, tensile strength at high temperatures, and outstanding corrosion and oxidation resistance [1,2]. However, the low machinability of Ni-based superalloys hampers their processing through traditional subtractive techniques. In addition, the hardened surface resulting from subtractive operations can lead to a reduction of cutting tool life, thus increasing the insert consumption [3,4]. All these complications have been overcome with the use of additive technologies. In particular, one of the most promising additive manufacturing technologies to process Ni-based superalloys is the laser-based powder bed fusion (PBF-LB/M) technique that allows the production of metallic complex-shaped parts guaranteeing high dimensional tolerance, low surface roughness, and thus reduced post-processing [5].

Among the Ni-based superalloys processable via the PBF-LB/M process, Inconel 625 (IN625) stands out for its impressive oxidation and corrosion properties up to around

1000 °C combined with good mechanical properties [6]. Although the IN625 samples processed via PBF-LB/M proved to result in densified and crack-free components [7,8], the high residual stresses induced by additive processing remain an unsolved problem [9]. The extremely high cooling rates achieved during the PBF-LB/M process ($10^5$ K s$^{-1}$) lead, in fact, to a strong thermal gradient between the melt pool and the previously consolidated cold layer [10]. This results in a high residual stress level in the final parts that could cause cracks, delamination, and distortion [11]. Moreover, residual stresses could be detrimental to mechanical performance, increasing the overall stresses applied to the component and thus leading to premature failure during static and dynamic loads [12,13]. For all these reasons, it is, therefore, crucial to quantify the residual stresses inside the components and study the possible strategies to mitigate them.

In the literature, several approaches are employed to estimate the residual stresses in IN625 components, ranging from micro-scale to macro-scale methods. Mishurova et al. performed residual stress measurements by X-ray diffraction analysis [14]. This approach allows the crystal lattice strain to be calculated and, thus, the residual stress level by considering a linear elastic distortion of the crystal lattice [15]. Lass et al. and Wang et al. measured the residual stress tensor averaged over a specified volume through neutron diffraction analysis [16,17]. With this method, it was possible to record slight changes in lattice spacing by high-resolution neutron diffraction [18]. Both X-ray and neutron diffraction analyses can be highly accurate but require expensive dedicated equipment and qualified operators. Another approach adopted in the literature is the hole-drilling method [19]. This method is known as semi-destructive, consisting of drilling a small hole in the sample and using a strain gauge to calculate the induced deformations. A software program will then match the measured deformations with the corresponding residual stress values [20]. However, some studies have questioned the reliability of the hole-drilling results [21], thus its use is recommended in combination with other investigative analyses. A further method to evaluate the residual stresses is the cantilever approach [9,22,23]. This approach involves the production of a cantilever with standard geometry that could deflect upon cutting from the building platform. The magnitude of beam deflection is directly proportional to the residual stresses created during the PBF-LB/M process. In fact, the amount of residual stresses directly influences the beam bending stiffness due to the geometry non-linearity and the effect of the Poisson ratio [24]. The cantilever approach is highly used to perform macro-scale comparative studies on residual stresses as it does not require advanced equipment or dedicated software and has a very short analysis time.

Looking more closely at mitigation strategies, some studies in the literature have emphasised the correlation between the process conditions of PBF-LB/M and the residual stresses generated in the end components [7,25,26]. Since scan speed and laser power had a high impact on the thermal gradient and liquid–solid interface velocity of the melt pool, it has been observed that high power and low scan speed values may contribute to reducing residual stress formation [7,26]. In addition, a building platform preheating is often performed to slow down the heat flow and thus reduce the thermal-induced stresses [25–27]. Another successful alternative strategy to reduce the heat flow is using support structures between the cold building platform and the component to be built [14,28–31].

Although the previously mentioned methods have proven to be effective in limiting residual stresses, a post-processing heat treatment as stress-relieving (SR) is recommended to avoid distortion after sample–platform detachment. A suitable stress-relieving treatment for IN625 processed by traditional technologies has already been developed, but the microstructural peculiarities of the PBF-LB/M process make the study of a tailored SR an ongoing task [8,16,32,33]. In fact, the traditional stress-relieving at 870 °C for 1 h [34], if performed on PBFed samples, proved to cause the accelerated formation of δ (Ni$_3$Nb) phases [16,33]. It is actually well known that a high concentration of coarse δ phases can cause a significant decrease in fracture toughness, ductility, and corrosion resistance [33–36]. For these reasons, it is crucial to investigate tailored stress-relieving heat treatments to reduce residual stresses avoiding undesirable δ phase formation. For the PBFed IN625

alloy, stress-relieving at 800 °C was proposed in order to reduce the concentration of δ phases [16,33].

To the best of the author's knowledge, no study has yet been conducted in the literature on low-temperature stress-relieving treatments that simultaneously investigate the effect of tailored SR treatments on residual stress reduction, microstructural characteristics, and mechanical performance. The aim of the present work is to find a proper stress-relieving temperature to balance residual stress relief, microstructural properties, and mechanical performance of an IN625 alloy processed by PBF-LB/M. The effectiveness of stress-relieving treatments was investigated by quantifying residual stresses through a synergic use of two approaches, the cantilevers approach as the macro-scale method and X-ray technology as the micro-scale method. In addition, the impact of SR temperatures on the microstructural features was evaluated via scanning electron microscope (SEM) investigations, electron backscattered diffraction (EBSD) maps, X-ray diffraction (XRD) patterns, and differential scanning calorimetry (DSC) analyses. Lastly, the mechanical behaviour in terms of microhardness and tensile properties was evaluated.

## 2. Materials and Methods

### 2.1. Powder Characterisation

A commercial gas atomised IN625 powder provided by EOS GmbH (Krailling/Munich, Germany) was used for the present study. The powder composition reported in Table 1 was evaluated through an energy dispersive spectroscopy (EDS) investigation for the main chemical elements and a Leco ONH 836 analyser (LECO Corporation, St. Joseph, MI, USA) for the measurement of the carbon concentration. The powder composition resulted in line with the UNS N06625 standard.

**Table 1.** IN625 chemical composition.

| Element (wt.%) | Ni | Cr | Mo | Nb | Fe | Co | Si | Al | Ti | C |
|---|---|---|---|---|---|---|---|---|---|---|
| IN625 | 65.8 | 20.5 | 8.1 | 3.9 | 0.7 | 0.1 | 0.3 | 0.3 | 0.3 | 0.012 |

The powder morphology was observed through an SEM analysis using Zeiss EVO 15 (Carl Zeiss AG, Jena, Germany). As illustrated in the SEM micrograph (Figure 1), the powder is characterised by a spherical shape with a limited presence of satellites and irregular particles according to the requirements of the PBF-LB/M process [37].

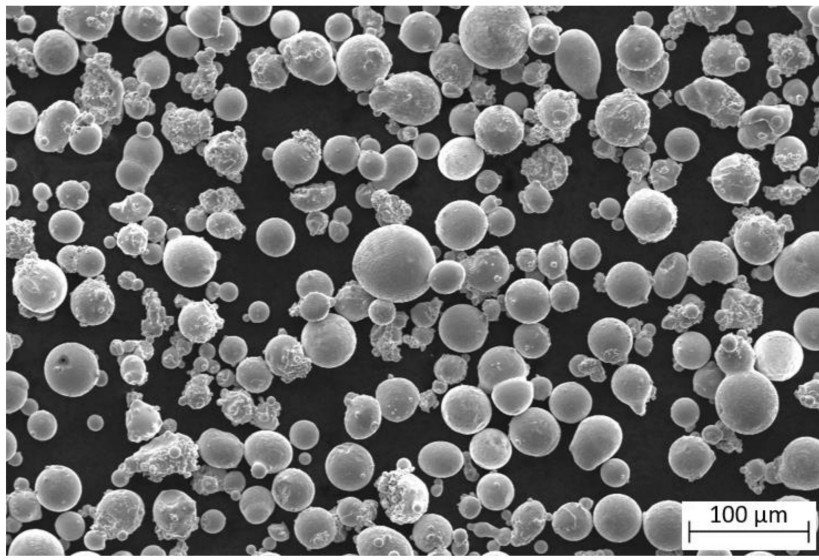

**Figure 1.** IN625 powder morphology observed via SEM.

### 2.2. PBF-LB/M Production and SR Treatments

The commercial IN625 powder was processed under a high-purity Argon atmosphere by the PBF-LB/M technique. The entire PBF-LB/M production was achieved with a Concept Laser Mlab (GE Additive, Boston, MA, USA) cusing R machine equipped with a 100 W fibre laser, a wavelength of 1070 nm, a laser spot of 50 μm, and $90 \times 90 \times 80$ mm$^3$ as build volume. The platform used for bulk production is 316L. As reported in the literature, the substrate on which the component is processed is relevant for the temperature distribution and thus for residual stresses [38,39]. Thanks to its limited cost, low thermal conductivity, and thermal expansion coefficient similar to the IN625 alloy, the 316L appears to be a qualified candidate as a building platform material to avoid increasing stress and save costs [38,39]. In order to obtain final parts with an optimum densification level, the parameters previously optimised by our research group were used, including a laser power of 95 W, a scanning speed of 1200 mm/s, a hatch distance of 0.04 mm, and a layer thickness of 0.02 mm [40]. A scanning strategy with stripes with 67° counterclockwise rotations between consecutive layers was adopted for bulk production. The layer-by-layer rotation of scan vectors results in isotropic stress tensors in contrast to the classical unidirectional scanning strategies [41,42]. In order to carry out a complete characterisation, three kinds of specimens were produced: 10 mm cubes for the evaluation of residual stresses by X-ray analysis and for microstructural investigations, cantilevers with the geometry illustrated in Figure 2 for the evaluation of residual stresses and flat tensile specimens designed according to ASTM E8 for the tensile test and built-in shape without post-processing operations. In addition, as reported in Figure 2, cantilever and flat specimens were orientated at 45° parallel to the building platform, as this orientation leads to a lower stress level [22,42].

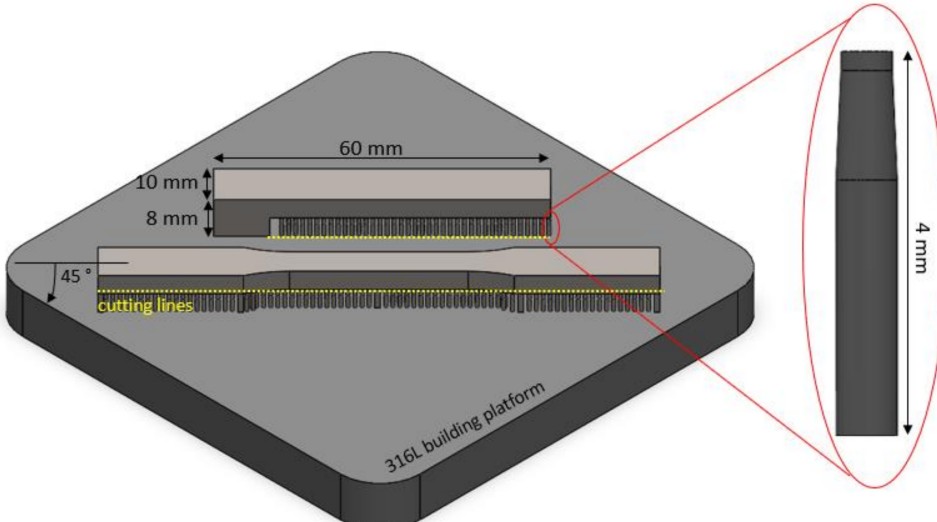

**Figure 2.** Specimen position on the building platform, geometry of support structures, and cutting lines followed.

Cylindrical supports with a truncated conical final part were chosen as support structures for the cantilevers and flat tensile specimen production. This particular shape guarantees a better heat flow between the component and the building platform [42]. The dimensions of the structures and their spacing were set to minimise the thermal gradient during the PBF-LB/M process while ensuring an adequate platform–sample adhesion [43]. After PBF-LB/M production, four stress-relieving temperatures were explored: 870 (the standard temperature), 800, 750, and 700 °C for a holding time of 1 h followed by air cooling. The stress-relieving treatments were performed in a muffle furnace using a heating rate of 5 °C/min to reach the desired temperature. After performing the desired stress-relieving treatment, all specimens were removed from the platform using an electrical discharge machine (EDM). The cutting lines are highlighted in yellow in Figure 2. Looking at the

cantilever samples, a partial cut from the platform was performed, leaving the bulk part attached to the platform and cutting the part with the supports. In this way, the cantilever free-end is free to deflect releasing residual stresses.

*2.3. Characterisations*

After performing the required EDM cuts on cantilever specimens, a Mahr XC20 coordinate measuring machine was used to measure the displacement of the cantilever free-end due to the stress relaxation. In order to have an adequate statistic, the free-end cantilever displacement to assess the effectiveness of SR treatments on tension relaxation was evaluated on three cantilevers for each condition. Residual stresses were also assessed on the specimen surface parallel to the build direction through Pulstec μ360x operated at 30 kV with a manganese cathode. The latter is a non-destructive technique based on the X-ray radiation of a polycrystalline solid and the measurement of the angles at which the highest intensities of diffracted X-rays occur. Using Bragg's law (Equation (1)) [15] the angle measurements could be correlated with the interplanar spacing (d) of the diffraction planes:

$$2d_{hkl} \sin \vartheta = n\lambda, \tag{1}$$

where n is the order of reflection, λ is the wavelength of the incident X-rays, and $\vartheta$ is the angle of incidence.

If residual stresses are present within the sample, d would be different from that of an un-stressed sample ($d_0$). In particular, to ensure the reliability of the results, the equipment was calibrated with a ferrite sample with a known residual stress level of 0 MPa. The difference between d and $d_0$ is proportional to the amount of residual stresses present in the component following Hooke's law [15] reported below:

$$\sigma = \frac{E}{(1+\nu) \sin 2\vartheta} \left( \frac{d_\vartheta - d_0}{d_0} \right), \tag{2}$$

where σ is the residual stress perpendicular to the measurement direction, E is the Young's modulus, ν the Poisson's ratio, and $d_\vartheta$ e interplanar spacing referred to $\vartheta$ angle.

This technique allows the measurement of residual stresses at the sample surface within a layer of 30–50 μm thickness. Since it is a surface technique, the measurements can be influenced by the surface roughness of the sample and the contour parameters used. However, considering that all samples were produced with the same process parameters, it is reasonable to assume that the measurements are comparable and the trend can provide useful information. In addition, to have an adequate statistic and to avoid the influence of the measurement positioning, the residual stresses were analysed at five points along the entire sample height.

After evaluating the residual stress for each condition with the synergic approach macro and micro-scale, a microstructural investigation was performed. All cubic samples, both as-built and SR-treated, were cut along the building direction and polished until Silica 0.05 μm. The samples were observed using an SEM EVO 15 microscope after electrolytic etching with orthophosphoric acid. Moreover, the grains were analysed by a Tescan S9000G FIB SEM (TESCAN, Brno, Czech Republic) equipped with an EBSD detector. EBSD orientation maps were recorded at 300× magnification and a step size of around 1–3 μm. The samples were tilted 70° and the SEM operated at 20 kV and 10 nA.

For phase identification and lattice parameter assessment, XRD measurements were performed on a PANalytical X-Pert diffractometer (Malvern Panalytical, Malvern, UK) using a Cu Kα radiation at 40 kV and 40 mA in a Bragg–Brentano configuration on the mid-planes of the samples parallel to the direction of construction. The general diffraction patterns between a 2 $\vartheta$ angle of 30° and 100° were recorded using a step size of 0.013° and a detection time of 30 s. In addition, a detailed XRD analysis was carried out on the first peak (42°–45°) with a step size of 0.003° and a detection time of 60 s.

The mechanical behaviour was investigated through Vickers hardness and tensile tests. A Vickers tester VMHT (Vickers, London, UK) was used to conduct the hardness testing according to ASTM E384 standards on the mid-planes of the samples parallel to the direction of construction. Ten hardness indentations were performed on the XZ plane using a static load of 0.5 kg and a dwell time of 15 s. The tensile test was carried out at room temperature using a Zwick Z100 tensile machine (ZwickRoell GmbH & Co., Ulm, Germany) and applying a strain rate of $8 \times 10^{-3}$ s$^{-1}$. During the tensile tests, a loading axis perpendicular to the build direction (BD) of the specimens was chosen, as it represents the most adverse loading condition. The strain was measured by a strain gauge with an initial length of 10 mm. In order to calculate the deformation values from specimen regions subjected only to uniaxial stress, the strain gauge was positioned at the middle of the gauge length. Three different samples for each condition were tested.

## 3. Results

### 3.1. Residual Stress Evaluation

A stress-relieving heat treatment is recommended for PBFed IN625 samples to mitigate high residual stresses that could compromise the mechanical properties of the final components. To verify the effectiveness of SR treatments in mitigating thermally induced stresses in the PBF-LB/M process, the cantilever deflection approach was first used. This technique is based on measuring a macroscopic distortion of the cantilever after its separation from the building platform [23,44]. More specifically, the stresses released by the material after EDM cutting are converted into plastic deformations deflecting the cantilever free-end. Generally, the greater the deflection, the greater the residual stresses accumulated by the component during the PBF-LB/M process. The maximum deflection values measured for the as-built and each SR condition are displayed in Figure 3.

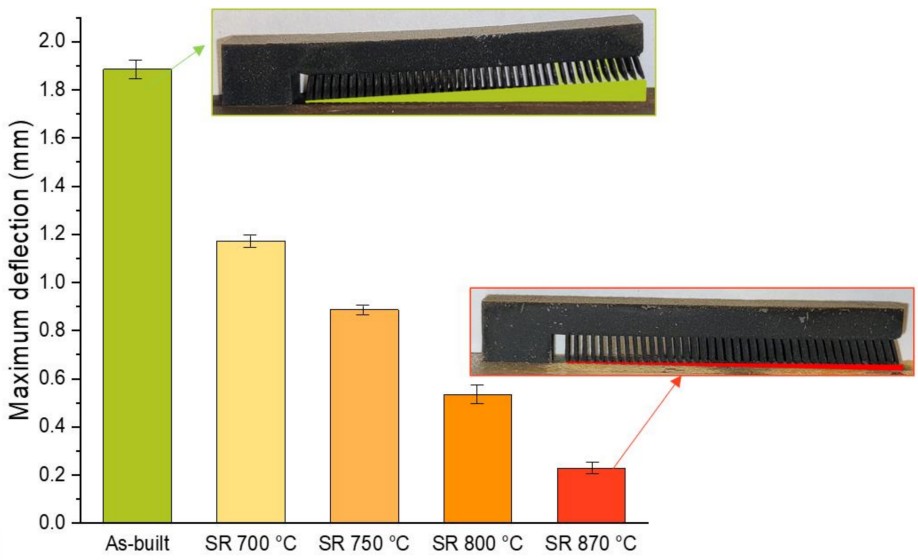

**Figure 3.** Cantilever maximum deflection for as-built and different SR conditions.

As can be observed in Figure 3, the increase in SR temperature leads to a progressive reduction of the measured distortion. Going into the details, the free end of the as-built cantilever underwent a mean maximum displacement of 1.88 mm. On the other hand, the stress-relieving treatments reduced the deflection from 38 to 88% as the treating temperature increased.

As a micro-scale approach, the X-ray method was then exploited. Examining samples treated at different SR temperatures, the obtained results are reported in Figure 4.

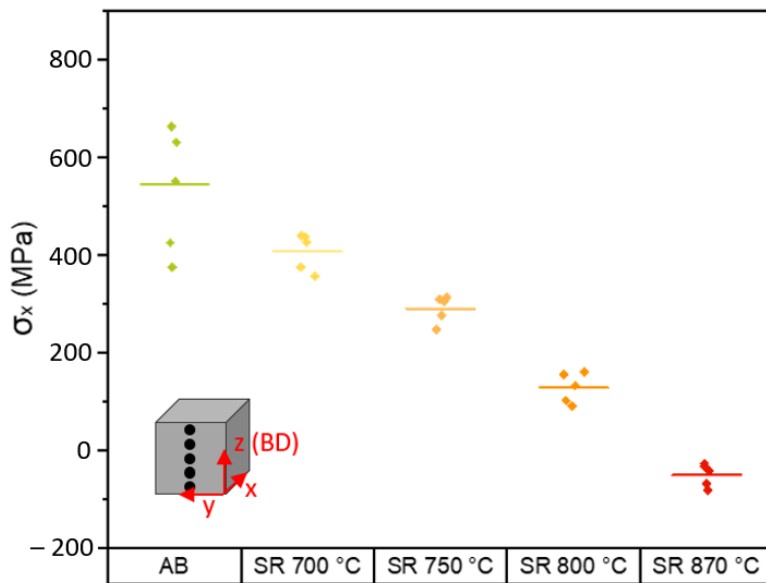

**Figure 4.** Residual stresses measured by high-energy X-ray diffraction in as-built and stress-relieved samples.

In the as-built state, a mean residual tensile stress value of 546 MPa with a standard deviation of 146 MPa was recorded. This average value is in line with the surface residual stresses recorded by other studies already published on IN625 processed for PBF-LB/M [7,16,17,45]. However, this high value of residual stresses in the as-built state could be dangerously close to the yield stress (YS) value of IN625 [16,33]. As stated by Chen et al. [46], if the thermally induced stresses exceed the YS of the material, the dislocation slip starts within the grains and could result in detrimental crack formation. As it is possible to observe from Figure 4, by performing the stress-relieving heat treatment, a significant decrease in residual stresses was detected. In particular, according to the macro-scale approach results, the higher the SR temperature, the lower the tensional state of the sample. Varying the SR temperature ranging from 700 to 870 °C, the mean residual stress value dropped from 387 to −55 MPa. During the PBF-LB/M process, the shallow surface undergoes repeated heating–cooling cycles that induce tensile stresses, while compressive stresses characterise the core [47]. The compressive nature of residual stresses recorded in the sample treated at the maximum SR temperature suggests that, at least at the surface level, the sample is in a favourable state of tension and this is in line with the results obtained by Barros et al. on stress-relieved IN718 samples through the hole-drilling strain brgauge method [48]. It is reasonable to assume that the SR at 870 °C released the superficial tensile stresses to a slightly compressive state as the core. The high standard deviation in the as-built condition could be attributed to the significant differences in the stress state as the layer-to-platform distance increases caused by the PBF-LB/M layer-by-layer processing. The homogenisation of residual stresses along the height of the sample is in line with that reported by Lass et al. [16].

The two investigative methods agreed, demonstrating the effectiveness of stress-relieving treatments in reducing residual stresses. In particular, both methods established that temperatures from 750 °C result in a reduction ranging from 50 to 90%. To investigate the influence of SR temperature on microstructural features and precipitation reactions, a microstructural investigation was carried out.

### 3.2. Effect of the Stress-Relieving Temperatures on Microstructural Features and Phase Formation

As a preliminary investigation of microstructure and spatial distribution of alloying elements of as-built and stress-relieved samples, an SEM investigation was conducted

with a backscattered electron detector. The SEM micrographs recorded for each process condition in the plane XZ (parallel to the BD) are shown in Figure 5.

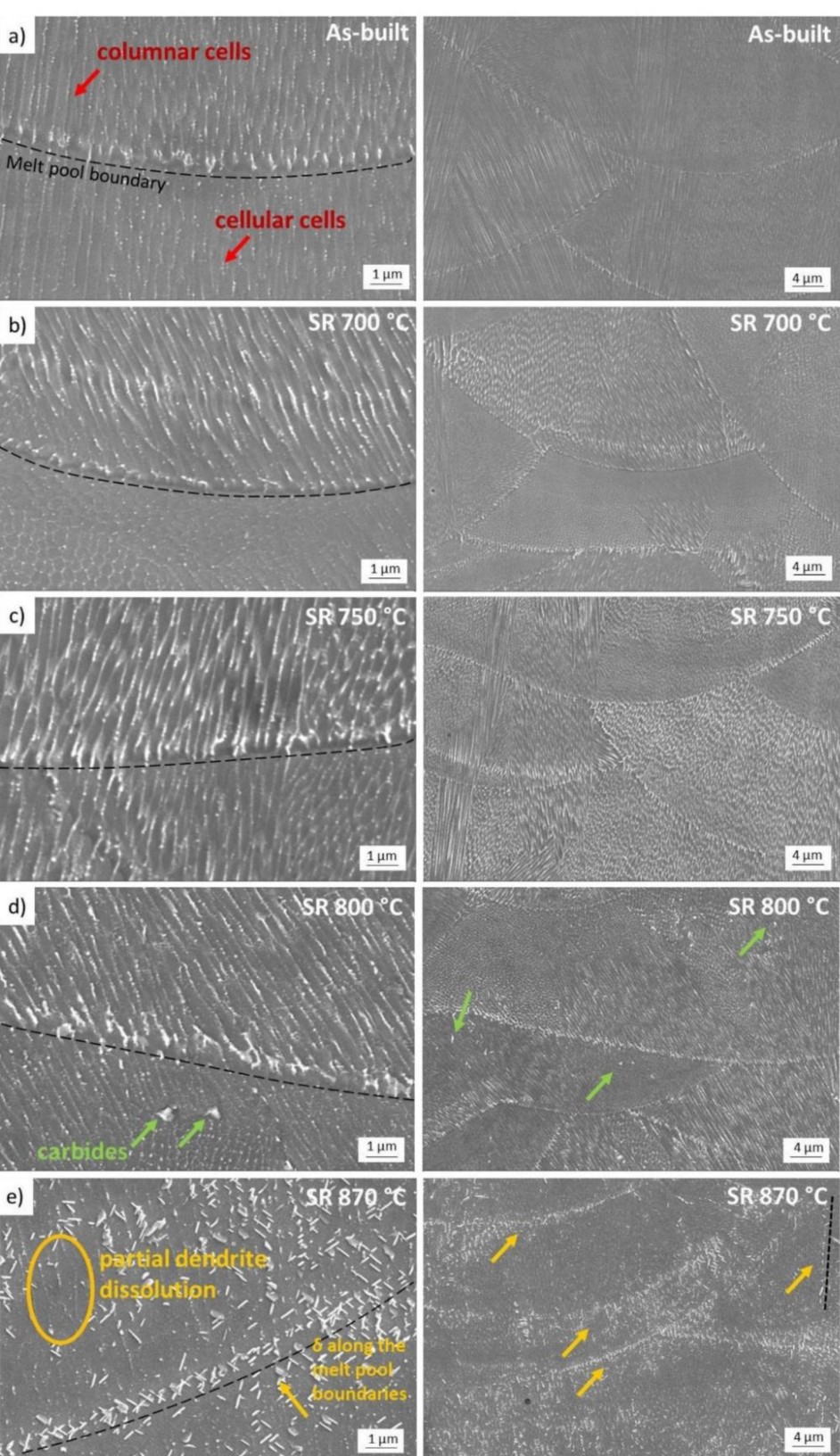

**Figure 5.** High (left images) and low magnification (right images) SEM micrographs for as-built condition (**a**) and for each stress-relieved state analysed (**b**–**e**).

In the as-built state, a segregated dendritic microstructure could be easily observed. The dendrites identified columnar and cellular cells (as indicated with red arrows in Figure 5a). The micro-segregations of chemical elements could be related to the rejection and redistribution of solute elements during the rapid solidification that occurred during PBF-LB/M production, as reported by a previous work of some of the authors [35]. By performing the stress-relieving treatments with increasing temperatures, the melt pool boundaries gradually became less visible, but their complete dissolution did not occur at the SR temperatures investigated (Figure 5b–e). Although the Time-Temperature-Transformation diagram of the alloys and the accelerated formation of phases commonly observed for the IN625 processed by PBF-LB/M due to the presence of micro-segregations should suggest the presence of $\gamma''$ phases [33], this was not detected with the SEM analyses carried out in the present work. According to the literature [35], $\gamma''$ phase should form with low-temperature SR treatment in nanometric form, making its detection only possible through an accurate TEM analysis. Some globular precipitates were observed in the SR 800 °C condition, where they appeared coarse (as indicated with green arrows in Figure 5d). Considering their shape and size, these precipitates may be carbides precipitated under thermal exposure [35]. The columnar and cellular cells identified by the dendrites remain clearly visible, performing the SR treatment until 800 °C. On the contrary, by performing the SR treatment at 870 °C an initial dissolution of the dendritic structures occurs, and the columnar and cellular cells appear less visible (as observable in the yellow circle in Figure 5e). In addition, at 870 °C the formation of the δ phase predominantly occurs along melt pool boundaries, grain boundaries (as indicated by the dotted line), and interdendritic areas where micro-segregations of Nb are present (as indicated in yellow in Figure 5e).

Performing the EBSD analysis a microstructural and crystallographic characterisation of PBF-LB/M IN625 was obtained. Figure 6 revealed an overall grain alignment along the BD, in line with the directional heat flow characterising the PBF-LB/M build process. This grain orientation remains almost unaltered for all stress-relieving conditions analysed, demonstrating that such treatments have a negligible effect on recrystallisation. From the EBSD maps, grain sizes were analysed as a function of the equivalent circular diameter (ECD) index. The frequency reported in Figure 6 is a volumetric frequency. Equiaxial grains represent a minority and represent an average ECD value of around 10 μm for each process condition. Looking at the size of the columnar grains, there is no significant growth as the SR temperature increases. Considering the data deviation, the peak shift towards higher ECD and increase in bigger grains fraction observable from Figure 6 are not statistically relevant. It is reasonable to assume that the δ precipitation does not allow grain growth. In fact, as stated by Li et al. the recrystallisation in IN625 PBFed samples occurs around 1000 °C [49].

In order to verify the effect of SR temperatures on the precipitation reactions in the PBF-LB/M samples of IN625, DSC analyses were conducted (Figure 7). In particular, the exothermic peak found between 420 and 650 °C (as minimum onset and maximum offset values) indicates the nucleation and growth of precipitates. The enthalpy released in exothermal processes can be measured as the area under the peak. The large area subtended by the DSC curve for the sample in the as-built condition denotes a high level of the solid solution generated during the PBF-LB/M process. Consequently, the decrease in enthalpy observable for the stress-relieved samples validates the precipitate formation hypothesis stated during the SEM investigations. In fact, although it was not possible to identify the $\gamma''$ phase with the SEM analysis, based on the time–temperature–transformation diagram of the alloys and the accelerated formation of phases commonly observed for the IN625 processed by PBF-LB/M due to the presence of micro-segregations [33], it is possible to assume that this phase is formed during low-temperature SR treatment. Based on the lower peak magnitude with respect to the as-built state, it is thus possible to infer that, during the SR from 700 to 800 °C, the thermal exposures already induce a partial formation of $\gamma''$ phases inside the material. The further decrease in the enthalpy for the 870 °C condition

can stem from the formation of micrometric δ phases, as already displayed in the SEM micrographs (see Figure 5e).

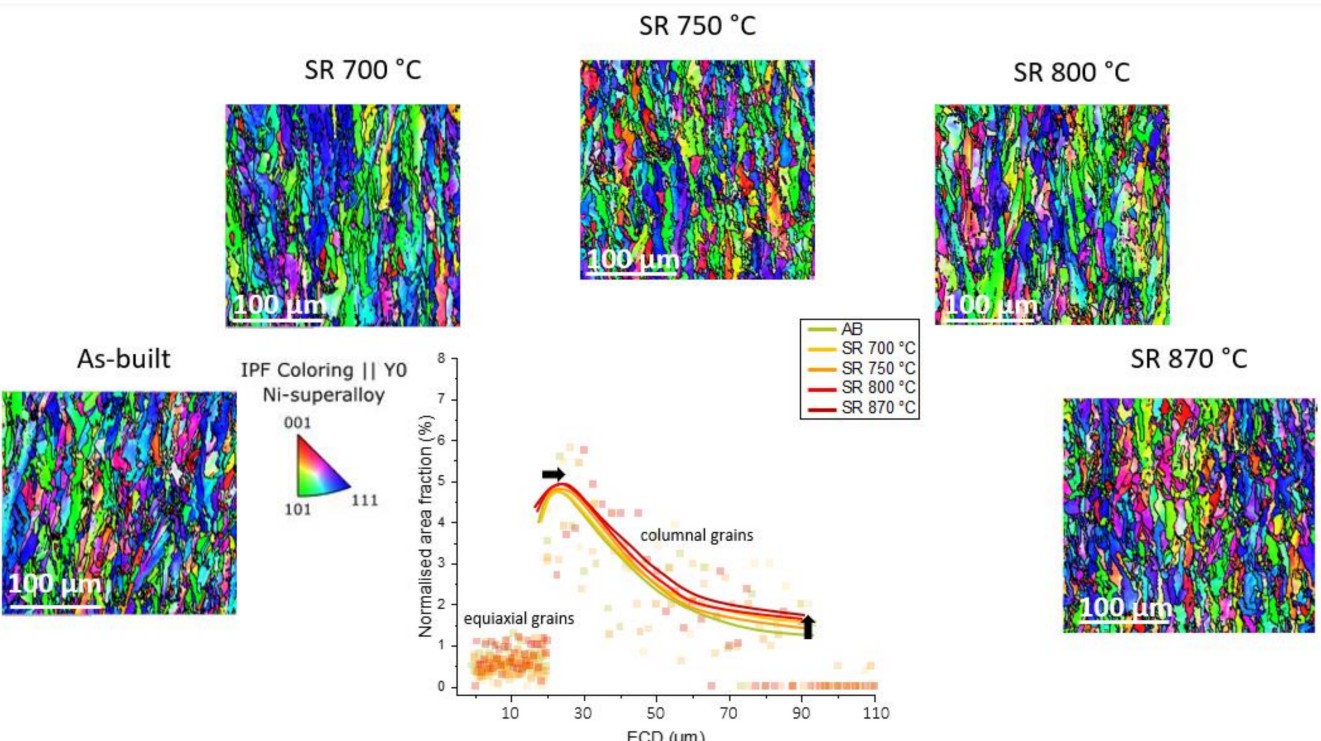

**Figure 6.** The EBSD orientation maps and grain size distribution before and after different stress-relieving treatments.

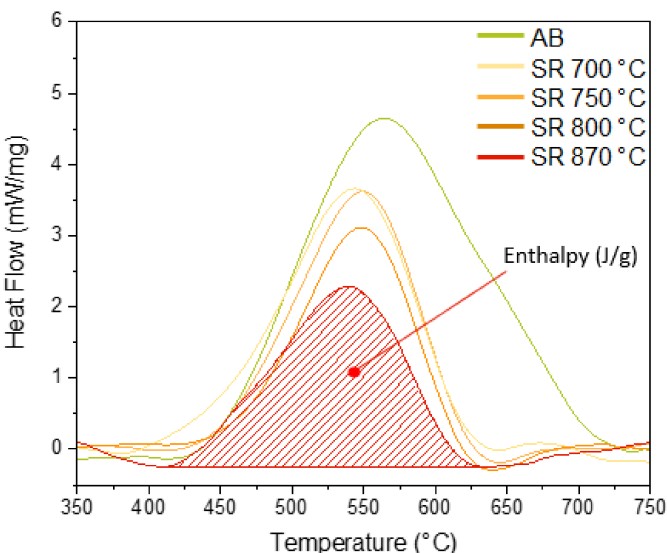

**Figure 7.** DSC signals of as-built and stress-relieved samples of IN625.

In order to perform phase identification and lattice parameter determination, XRD analyses were performed on XZ cross-sections of specimens in the built and stress-relieved state. All the collected X-ray diffraction patterns are compliant with the typical γ-Ni matrix characterised by a face-centred cubic (FCC) lattice, as shown in Figure 8.

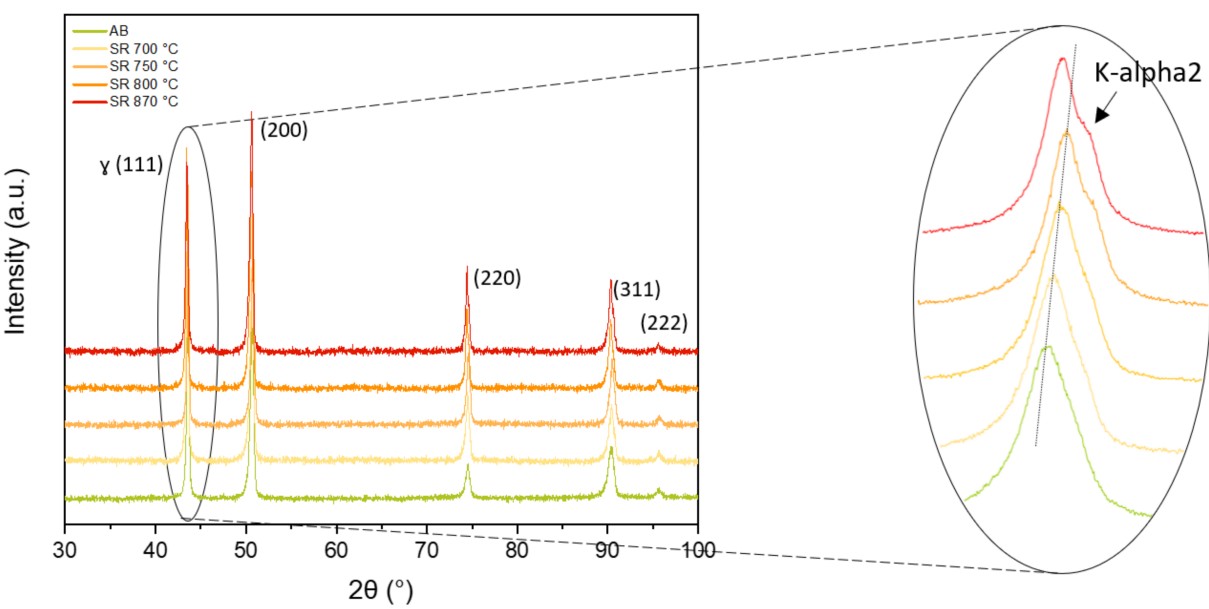

**Figure 8.** The XRD patterns of the as-built and stress-relieved IN625 samples.

Carbides or intermetallic phases, such as δ phase which may be formed during heat treatment, were not detected by XRD analysis probably due to their limited size and the detection limit of the instrument. However, by looking more closely at the detailed XRD patterns of the first peak, it is possible to obtain information on the lattice parameter and crystallite size. A shift towards higher angles of the first peak by increasing the temperature of the SR from 700 to 800 °C is evident. Based on Bragg's law, this shift gives indirect evidence that increasing the SR temperature up to 800 °C involves a reduction of the lattice parameter. This reduction could be attributed not only to the initial release of residual stresses but also to the formation of the phases. In fact, the precipitation of phases rich in chemical elements such as Nb and Mo reduces their concentration within the matrix, causing a constriction of the lattice parameter. Differently, the peak position of the samples heat-treated at 870 °C shows a shift towards higher angles, thus indicating an increment of the lattice parameters. In fact, even if the formation of δ phases leads to a decrease in the lattice parameter due to the depletion of Nb and Ni, the initial dissolution of the dendritic structures may result in an increase in the lattice parameter. As established in previous work by the authors [35], dendrites are enriched in Nb and Mo. Therefore, by observing a partial dissolution of the dendrites as the temperature of SR increases (Figure 5e), it is reasonable to assume that Nb and Mo return to the matrix, causing an increase in the lattice parameter. The combination of these two phenomena results in an overall increment of the lattice parameter.

In addition, by observing the first peak shape, it is possible to state that by increasing the SR temperatures over 750 °C, k-α2 peaks appear more evident. This suggests a growth of crystallites which results in a narrowing of the peaks and, thus the appearance of the peak relative to k-α2.

### 3.3. Impact of the SR Treatments on the Mechanical Performance: Hardness and Tensile Test Results

Table 2 reports the mechanical properties related to the as-built and stress-relieved conditions obtained by Vickers hardness and tensile tests. Tensile proprieties are expressed as YS, ultimate tensile stress (UTS), and elongation at the break (%A).

**Table 2.** YS, UTS, and %A values with relative standard deviation for each process condition based on tensile test results.

|  | **AB** | **SR 700 °C** | **SR 750 °C** | **SR 800 °C** | **SR 870 °C** |
|---|---|---|---|---|---|
| Hardness (HV) | 342 ± 7 | 360 ± 8 | 363 ± 7 | 361 ± 6 | 346 ± 4 |
| YS (MPa) | 566 ± 23 | 802 ± 17 | 829 ± 28 | 800 ± 24 | 733 ± 23 |
| UTS (MPa) | 1087 ± 16 | 1066 ± 11 | 1099 ± 14 | 1074 ± 15 | 1031 ± 5 |
| %A (%) | 29.3 ± 0.2 | 29.6 ± 1.2 | 29.6 ± 0.2 | 28.9 ± 1.1 | 33.4 ± 0.2 |

The mean hardness value of 342 HV recorded in as-built specimens is consistent with that presented by Li et al. [8]. The as-built specimens showed higher hardness with respect to the forged samples (approx. 305 HV) due to the finer microstructure typical for a PBF-LB/M material [8]. By performing the stress-relieving treatments between 700 and 800 °C, the average hardness values increased to around 365 HV. This increase in hardness may result from the finely dispersed precipitation of phases such as the $\gamma''$ phase [32]. On the contrary, the stress-relieved samples at 870 °C showed a decreased hardness value with respect to the other SR conditions. This could be explained by the two concurrent phenomena of δ precipitation and the initial dissolution of the dendritic structures. In fact, the hardening that the δ phase precipitation should bring is outweighed by the weakening caused by the initial dissolution of the dendritic structures observed with SEM investigations and supported by XRD analysis. The aforementioned results in a hardness value of the sample stress relived at 870 °C resulted to be similar to that in the as-built state.

As expected, the YS values revealed a trend similar to the hardness results. In particular, the stress-relieved samples up to 800 °C showed a significant increase in the YS with respect to the as-built condition (approximately 45%). This increase could be explained by the significant strengthening role played by the $\gamma''$ phase due to its large misfit with the $\gamma$ matrix. Even the stress-relieved sample at 870 °C exhibited an increase in YS compared to the as-built condition, but it was less intensive than the samples treated up to 800 °C because δ has a lower strengthening power than $\gamma''$. Particular attention must be paid to the elongation-at-break values. While no significant differences are noticeable for SR conditions between 700 and 800 °C considering the standard deviations, an increase in ductility occurs for the sample heat-treated at 870 °C. This increase contrasts with the well-known ductility loss caused by the δ phase but can be justified by the initial dendritic dissolution observed with SEM investigation (Figure 5e). In fact, as explained in the literature [6,50], dendrite dissolution increases plasticity while decreasing alloy strength.

The study of the impact of SR temperature on IN625 mechanical properties allowed further screening of the temperatures resulting from the analysis of residual stresses and microstructural features. In particular, the mechanical investigations identified 750 and 800 °C as the temperatures to obtain an increment of the YS combined with a reduction of residual stresses and a limited alteration of the dendritic structures.

## 4. Conclusions

The problem of thermally induced stresses caused by the extremely high PBF-LB/M cooling rates remains one of the most detrimental aspects of IN625 additive processing. Although appropriate optimisation of process conditions can decrease this adverse phenomenon, a post-processing stress-relieving treatment is always recommended in order to achieve a residual stress level compatible with deformation-free platform-to-component cutting. The stress-relieving procedure recommended by traditional wrought processed IN625 is reported as 870 °C for 1 h. This treatment mitigates the residual stresses but simultaneously involves the formation of the undesired δ phase. Exploring standard and low-temperature stress-relieving treatments by correlating residual stress mitigation, microstructural properties, and mechanical behaviour, the present work collected the results summarised below:

- The multiscale approach, cantilever, and X-ray methods, adopted to verify the effectiveness of stress-relieving treatments in stress mitigation, established that temperatures from 750 °C result in a residual stress reduction ranging from 50 to 90%.
- Microstructural analyses revealed that the columnar and cellular dendritic structures typical of the as-built state are preserved even after SR treatments up to 800 °C, showing only slight thickening of the cell walls and the formation of carbides. In addition, DSC and XRD analyses suggested the possible formation of $\gamma''$ phase for SR performed at temperatures under 800 °C. At the SR temperature of 870 °C, the microstructure significantly changed: the dendritic structure started to dissolve and the formation of the $\delta$ phase took place. The $\delta$ phase was needle-like and mainly distributed at the interdendritic areas, melt pool contours, and grain boundaries.
- The increase in mechanical properties (hardness and YS) obtained with the stress-relieving treatment up to 800 °C supports the initial phase formation in these SR conditions. However, the sample treated at 870 °C did not follow the trend and showed decreased mechanical proprieties and improved plasticity. Although $\delta$ formation in samples treated at 870 °C should have increased hardness and strength, the initial dissolution of dendritic structures led to a general decrease in mechanical behaviour and an increase in plasticity with respect to the other investigated SR conditions.

Considering that the primary goal of stress-relieving treatment should be to combine a reduction of thermally induced stresses with the least altered microstructure, the present work concluded that the two heat treatments at 750 and 800 °C for 1 h could represent alternative stress-relieving for the additively manufactured IN625 alloy. With these two stress-relieving treatments, it is possible to obtain a reduction above 50% of residual stresses, an almost unaltered microstructure, and improved mechanical behaviour. The present study established new stress-relieving temperatures for IN625 samples processed through PBF-LB/M by applying a wide-ranging approach to investigate the residual stress level, microstructure, and mechanical properties.

**Author Contributions:** Conceptualization, A.M., G.M. and M.L.; methodology, A.M., G.M. and E.B.; investigation, A.M. and E.B.; data curation, A.M.; writing—original draft preparation, A.M.; writing—review and editing, A.M., G.M. and E.B.; supervision, G.M. and M.L. All authors have read and agreed to the published version of the manuscript.

**Funding:** This research received no external funding.

**Conflicts of Interest:** The authors declare no conflict of interest.

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
