# Peer review of "Effects of Stress-Relieving Temperature on Residual Stresses, Microstructure and Mechanical Behaviour of Inconel 625 Processed by PBF-LB/M"

_metals, doi:10.3390/met13040796_

Round 1

Reviewer 1 Report

Effects of stress-relieving temperature on residual stresses…

2.2. PBF-LB/M

Lines 150-155: the description is not clear – draw testing specimens into Fig.2, as they were cut out.

2.3. Characterisations

Line 175: where (show in the Figure 2) were evaluated residual stresses with X-ray?

Equations 1 and 2: missing reference

Line 186: is ferrite sample OK for calibration a manganese cathode?

Line 208: XRD measurement (missing of what) were…   It must be explained, because the reader could be confused with explanation in line 178.

3.1. residual stress evaluation

Figure 4: it is not clear, where resiadual stresses were measured (show in Fig.2) – line 267 (characterise the core???)

Line 272: as the core? – in the core?

3.2. effect of the…

Line 346: XRD analyses (missing: of what) were…

3.3. Impact of the SR treatments…

Missing: how many hardness and tensile tests were made?

Line 378: HV0,5 is not micro-Vickers hardness. It is just hardness – correct every else.

Line 379: dL should be %A (I believe). Additional to table 2 you could add also s-e curves.

Line 398: a significant increment – a significant increase?

Line 399: increment – increase?

Author Response

Firstly, the authors wish to express their thanks for the time and effort the reviewer devoted to reading through the manuscript. Every suggestion has been an opportunity to implement and improve the manuscript quality. All comments were collected and analysed and the manuscript was amended accordingly. The updates have been reported in red in the manuscript.

The revisions received are analysed point by point below:

2.2. PBF-LB/M

Lines 150-155: the description is not clear – draw testing specimens into Fig.2, as they were cut out.

The authors thank the reviewer for the suggestion. The explanation of the geometry and positioning of the samples has been better detailed, and Figure 2 has been redrawn to better emphasise the positioning and cutting lines with which the samples were removed from the platform.

2.3. Characterisations

Line 175: where (show in the Figure 2) were evaluated residual stresses with X-ray?

In line 178, a specification was added on the location of the XRD measurements to detect residual surface tension. Furthermore, the site of the measurements is illustrated in Fig. 4.

Equations 1 and 2: missing reference

References have been added as requested.

Line 186: is ferrite sample OK for calibration a manganese cathode?

The machine calibration was carried out according to the machine supplier's recommendations.

Line 208: XRD measurement (missing of what) were…   It must be explained, because the reader could be confused with explanation in line 178.

This specification was added as requested.

3.1. residual stress evaluation

Figure 4: it is not clear, where resiadual stresses were measured (show in Fig.2) – line 267 (characterise the core???)

Line 272: as the core? – in the core?

According to the literature, samples produced for PBF-LB/M are characterised by a surface tensile state and a compressive state at the core. With surface treatment at 870 °C, the tensile stresses were completely released and a light compressive state was established (as the core).

3.2. effect of the…

Line 346: XRD analyses (missing: of what) were…

As requested by the reviewer, this part of the manuscript was revised.

3.3. Impact of the SR treatments…

Missing: how many hardness and tensile tests were made?

This information can be found in the materials and methods section.

Line 378: HV0,5 is not micro-Vickers hardness. It is just hardness – correct every else.

This correction was applied to the entire manuscript.

Line 379: dL should be %A (I believe). Additional to table 2 you could add also s-e curves.

The authors thank the reviewer for the suggestion, however, the curves were not included only because more tests were performed for each condition and it could be not very clear to the reviewer. The table is more straightforward to see not only the mean values but also the standard deviations. However, if required, they can be reported in the appendices.

Line 398: a significant increment – a significant increase?

Line 399: increment – increase?

Both corrections were implemented in the manuscript.

Reviewer 2 Report

The authors addressed a suitable stress-relieving temperature to balance residual stress relief, microstructural properties, and mechanical performance of an IN625 alloy processed by PBF-LB/M. The microstructure and residual stress were studied by conventional characterization and mechanical tests widely used in material science at different temperatures. The number of instrumental techniques is adequate to ascertain the main conclusions about the microstructure and residual stress and mechanical properties correlation. Notwithstanding, although the research topic is of interest to this journal, several issues were, however, not investigated in sufficient detail and should be improved.

- General remarks: The paper, the figures, and the general structure, as mentioned above, is complete and well-presented. The introduction and experimental section describe in detail the state-of-the-art and the methodology used. 

However, a key factor to consider for further improvement is the scientific discussion: 

- Results and discussion: 

Section 3.1: Please try to exclude descriptive paragraphs on techniques in this section. Move them to Introduction.

- Add error bars in figure 3 and detail the number of trials in the experimental part.

- “This average value is in line with the surface residual stresses recorded by other studies already published on IN625 processed for PBF-LB/M.”

Why? 

- Avoid repetition: 

“A stress-relieving heat treatment is recommended for PBFed IN625 samples to mitigate high residual stresses that could compromise the mechanical properties of the final components.”

“To avoid the crack formation and plastic deformation, stress-relieving heat treatment is recommended to relax these residual stresses.”

 - Section 3.2

- “The micro segregations of chemical elements could be related to the rejection and redistribution of solute elements during the rapid solidification that occurred during PBF-LB/M production, as reported by a previous work of some of the authors”

What elements? Please provide a more detailed microstructural characterization (what are the present phases, chemical composition and processing parameters-microstructure(composition relation). Besides, a benchmarking to its counterpart processed via conventional should be added.

For instance, in Section 3.2 “In fact, even if the formation of δ phases lead to a decrease in the lattice parameter due to the depletion of Nb and Ni, the initial dissolution of the dendritic structures brings Nb and Mo into the matrix increasing the lattice parameter.”

 This is quite a “daring” statement based on the lack of EDS analysis.

- Were the “γ’’ phases” detected in the SEM analysis?

- Please provide more references where the scientific statements were done (applicable to this section).

- “k-alpha2”. Please, put it in greek format (Kα2).

- Section 3.3 and conclusions: Please, adapt the scientific discussion when a compositional analysis is provided.

Author Response

Firstly, the authors wish to express their thanks for the time and effort the reviewer devoted to reading through the manuscript. Every suggestion has been an opportunity to implement and improve the manuscript quality. All comments were collected and analysed and the manuscript was amended accordingly. The updates have been reported in red in the manuscript.

The revisions received are analysed point by point below:

- General remarks: The paper, the figures, and the general structure, as mentioned above, is complete and well-presented. The introduction and experimental section describe in detail the state-of-the-art and the methodology used. 

The authors sincerely thank the reviewer for appreciating their work and, in particular, the development of the introduction section.

However, a key factor to consider for further improvement is the scientific discussion: 

- Results and discussion: 

Section 3.1: Please try to exclude descriptive paragraphs on techniques in this section. Move them to Introduction.

- Add error bars in figure 3 and detail the number of trials in the experimental part.

The authors apologise for the oversight. The error bars have been included in Fig. 3 and the number of samples on which measurements were made has been added in the materials and methods section.

- “This average value is in line with the surface residual stresses recorded by other studies already published on IN625 processed for PBF-LB/M.”

Why? 

Realising that the mean value of the residual surface stresses in the as-built state was very close to the YS value, we wanted to compare our values with others obtained in the literature. With the literature search, we verified that the magnitude of residual stresses is approximately similar (despite different parameters and process conditions)

- Avoid repetition: 

“A stress-relieving heat treatment is recommended for PBFed IN625 samples to mitigate high residual stresses that could compromise the mechanical properties of the final components.”

“To avoid the crack formation and plastic deformation, stress-relieving heat treatment is recommended to relax these residual stresses.”

The authors thank the reviewer for the suggestion. The repetition has been removed from the manuscript.

- Section 3.2

- “The micro segregations of chemical elements could be related to the rejection and redistribution of solute elements during the rapid solidification that occurred during PBF-LB/M production, as reported by a previous work of some of the authors”

What elements? Please provide a more detailed microstructural characterization (what are the present phases, chemical composition and processing parameters-microstructure (composition relation). Besides, a benchmarking to its counterpart processed via conventional should be added.

Thanks for the comment. In a previous published paper (Marchese, G. et al. Influence of heat treatments on microstructure evolution and mechanical properties of Inconel 625 processed by laser powder bed fusion. Mater. Sci. Eng. A 729, 64–75 (2018)), we studied the microstructure of the as-built condition. The chemical elements that tend to segregate are mainly Nb and Mo. These elements tend to remain inside the interdendritic areas during the solidification. Several papers on LPBFed IN625 reported the enrichment of Nb and Mo in the interdendritic areas during the solidification. However, in the present work, the focus is on proving the impact of SR temperature on the residual stress relieving and the microstructural and mechanical properties of the alloy. For this reason, an in-depth microstructural investigation in the as-built state has been deferred to the literature.

For instance, in Section 3.2 “In fact, even if the formation of δ phases lead to a decrease in the lattice parameter due to the depletion of Nb and Ni, the initial dissolution of the dendritic structures brings Nb and Mo into the matrix increasing the lattice parameter.”

 This is quite a “daring” statement based on the lack of EDS analysis.

The authors have rewritten this sentence in order to make it clearer. In previous work by the authors, it was established that dendrites in the as-built state were rich in Nb and Mo. Therefore, observing in Fig. 5e the partial dissolution of the dendrites, it is reasonable to assume that Nb and Mo return to the matrix and cause the increase in lattice parameter proven by XRD analysis. The authors agree with the reviewer and several sentences in the manuscript have been revised to clarify what is observed in the present work and what is assumed on the basis of the literature and on the evidence obtained from XRD, DSC etc.

- Were the “γ’’ phases” detected in the SEM analysis?

Thank you for your comment.

In the manuscript, we reported that the heat treatments provide the formation of precipitates. Based on the accelerated formation of phases in the PBF process and considering the TTT diagram of the alloy, it is possible to infer that the heat treatment involves the formation of gamma double prime phase.

The formation of phases is further supported by the remarkable increment of hardness and mechanical properties without modifying the grain size compared to the as-built condition. However, the sentence was modified to explain the formation of phases, probably gamma double prime.

- Please provide more references where the scientific statements were done (applicable to this section).

- “k-alpha2”. Please, put it in greek format (Kα2).

 This substitution was applied to the entire manuscript.

Reviewer 3 Report

line 156: Please add another technical drawing of the geometry. The figure 2 ( right side) is not clear what it represents.

line 185: Why was the calibration not performed on a nickel sample?

line 199: Please add a drawing of the measuring position for the X-ray measurement.

line 213: Which X-ray elastic constants were used to calculate the stress?

line 217:  Please add a figure showing the position of the XZ-plane.

line 202: How many samples were tested in the tensile test?

line 286: plan XZ --> plane XZ

line 343: Please select other colours in figure 7 to better see the differences in the curve.

Author Response

Firstly, the authors wish to express their thanks for the time and effort the reviewer devoted to reading through the manuscript. Every suggestion has been an opportunity to implement and improve the manuscript quality. All comments were collected and analysed and the manuscript was amended accordingly. The updates have been reported in red in the manuscript.

The revisions received are analysed point by point below:

line 156: Please add another technical drawing of the geometry. The figure 2 ( right side) is not clear what it represents.

The description and Fig. 2 have been revised to better emphasise the sample positioning and cutting directions.

line 185: Why was the calibration not performed on a nickel sample?

The machine calibration was carried out according to the machine supplier's recommendations.

line 199: Please add a drawing of the measuring position for the X-ray measurement.

The authors are thankful for the suggestion. The sample image with the measurement placements highlighted is shown in Fig. 4.

line 213: Which X-ray elastic constants were used to calculate the stress?

For the determination of residual stresses a constant elastic modulus of 212 +\- 2 GPa and a poisson ratio of 0.31 +|- 0.01 was used as reference for un-stressed pure nickel

line 217:  Please add a figure showing the position of the XZ-plane.

The x, y and z directions of the cubic samples are now shown in Fig. 4.

line 202: How many samples were tested in the tensile test?

This information is reported in Material and Methods section (line 230).

line 286: plan XZ --> plane XZ

The authors sincerely apologise for the oversight.

line 343: Please select other colours in figure 7 to better see the differences in the curve.

The authors opted to use the same colour code for the entire paper in order to avoid confusing the readers. However, according to the reviewer, the thickness was increased to make the lines more visible.

Round 2

Reviewer 1 Report

line 391: micro-Vickers - change it to Vickers

Table 2: Is DL (%) = %A  ? Please add explanation or change designation.

%A=DL/L‧100%

Author Response

The authors apologise for the oversight in line 391. Furthermore, as the reviewer suggested, it was decided to rename elongation at the break as %A (where %A is DL/L‧100%). The authors express their sincere thanks for the suggested corrections that led to an improvement of the manuscript. 

Reviewer 2 Report

The authors have correctly reviewed and addressed all the initial issues and comments. Now, from my perspective, the article meets the quality requirements for publication in the metals journal.

Congratulations.

Author Response

The authors sincerely thank the reviewer for recognising the quality of the present work. Every correction was extremely helpful for the improvement of the manuscript. 

Thank you again.